# VLANeXt: Recipes for Building Strong VLA Models

Xiao-Ming Wu [1]  Bin Fan [2]  Kang Liao [1]  Jian-jian Jiang [2]  Runze Yang [3]  Yihang Luo [1]  Zhonghua Wu [4]
Wei-Shi Zheng [2]  Chen Change Loy [1 5]

https://dravenalg.github.io/VLANeXt/

## Abstract

Following the rise of large foundation models, Vision–Language–Action models (VLAs) emerged, leveraging strong visual and language understanding from Vision-Language Models for general-purpose policy learning. Yet, the current VLA landscape remains fragmented and exploratory. Although many groups have proposed their own VLA models, inconsistencies in training protocols and evaluation settings make it difficult to identify which design choices truly matter. To bring structure to this evolving space, we reexamine the VLA design space under a unified framework and evaluation setup. Starting from a simple VLA baseline similar to RT-2, which is the origin of VLA, we systematically dissect design choices along three dimensions: foundational components, perception essentials, and action modelling perspectives. From this study, we distill 12 key findings that together form a *practical recipe* for building strong VLA models. The outcome of this exploration is a simple yet effective model, **VLANeXt**. It outperforms the state-of-the-art methods on the LIBERO and LIBERO-plus benchmarks and demonstrates strong performance in real-world experiments. We release a unified and easy-to-use codebase to reproduce our findings, explore the design space, and develop new VLA variants on top of a shared foundation. The codebase is available at https://github.com/DravenALG/VLANeXt.

## 1. Introduction

Recent advances in foundation models have reshaped how we think about general-purpose robot control. Instead of

[1]S-Lab, Nanyang Technological University [2]Sun Yat-sen University [3]Shanghai Jiao Tong University [4]SenseTime Research [5]ACE Robotics. Correspondence to: Chen Change Loy <ccloy@ntu.edu.sg>.

*Proceedings of the 43rd International Conference on Machine Learning*, Seoul, South Korea. PMLR 306, 2026. Copyright 2026 by the author(s).

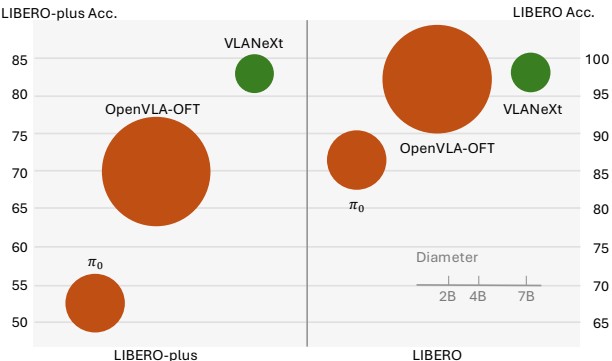

Figure 1. **Performance comparison on the LIBERO and LIBERO-plus benchmarks**. We compare VLANeXt with representative VLA baselines across model scales. Despite its smaller model size, VLANeXt achieves higher success rates than prior methods on both standard task performance (LIBERO) and robustness/generalization (LIBERO-plus), demonstrating the effectiveness of the design recipe distilled in this work.

training task-specific policies, a growing line of work builds Vision–Language–Action (VLA) models that leverage large Vision-Language Models to map visual observations and language instructions directly to robot actions. By inheriting rich visual understanding and language grounding from foundation models, VLAs[1] offer a scalable route toward general-purpose, language-conditioned robot policies (Ma et al., 2024; Ravichandar et al., 2020; Xiao et al., 2025c).

Since the emergence of VLAs (Zitkovich et al., 2023), both academia and industry have proposed a wide range of models demonstrating strong performance and encouraging generalization across diverse tasks (Zitkovich et al., 2023; O'Neill et al., 2024; Li et al., 2023b; 2024; Kim et al., 2024a; Black et al., 2024; Team et al., 2025; Hung et al., 2025; Kim et al., 2025; Shukor et al., 2025; Intelligence et al., 2025b;a; Liu et al., 2026). Most VLA approaches build on a similar paradigm: they build on pre-trained LLMs or VLMs, processing visual observations together with language instructions to derive action-relevant representations for policy learning. This pipeline introduces numerous design choices, including how to interface the VLM with the policy module, how to train the policy, how to select essential perceptual

[1]An overview of VLAs is provided in the Appendix.

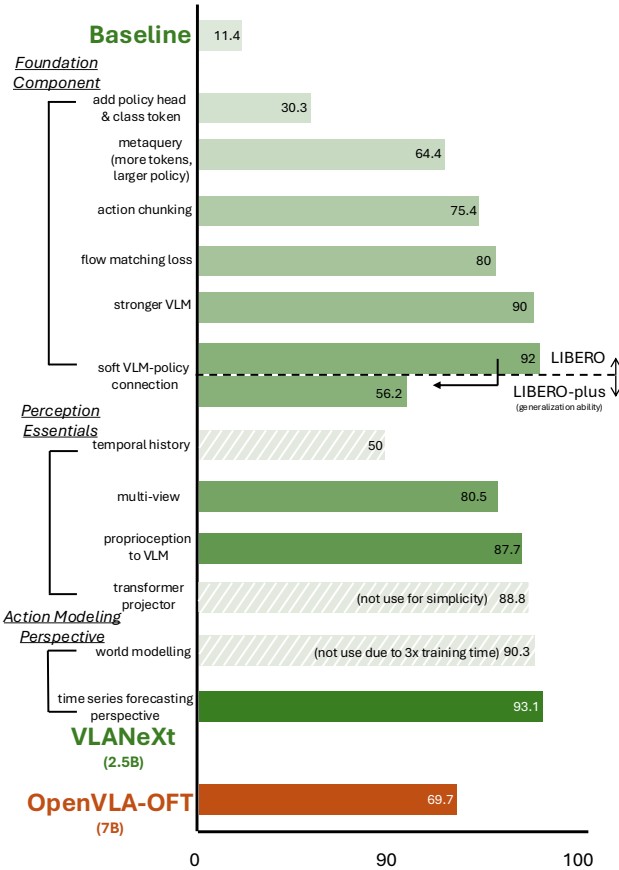

*Figure 2.* **Ablation trajectory across the VLA design space (spatial suite).** We progressively evolve a baseline VLA through changes in foundational components, perception, and action modeling. Results are reported on LIBERO initially, and on LIBERO-plus once LIBERO performance saturates, providing a more sensitive test of robustness and generalization. The trajectory culminates in the final VLANeXt model (2.5B) vs. OpenVLA-OFT (7B).

inputs, and how actions should be represented and modeled. Despite rapid progress, early exploration of VLAs remains something of a "primordial soup", rich in ideas but lacking clear structure. While prior work has explored VLA design from certain perspectives (Zhen et al., 2024; Qu et al., 2025; Zhang et al., 2025c; Cen et al., 2025b; Zhang et al., 2025b;d; Lu et al., 2025), differences in training protocols and evaluation setups make it difficult to identify which design choices in the shared VLA design space truly matter.

This work aims to provide a more systematic understanding of this fragmented design space by comprehensively reexamining VLA design spaces under a unified framework and evaluation protocol. While several prior works (Kim et al., 2025; Liu et al., 2026) have made preliminary attempts to explore VLA designs, their investigations remain limited in scope. This study aims to provide a more comprehensive and in-depth analysis of this domain. In detail, we begin with a simple baseline VLA, similar to RT-2 (Zitkovich et al., 2023), which is the origin of VLA and serves as a

strong reference point for analyzing the effectiveness of different design choices. We evaluate all variants on two commonly used VLA benchmarks, including LIBERO (Liu et al., 2023a) and LIBERO-plus (Fei et al., 2025b), where LIBERO-plus extends the original benchmark with controlled and unseen perturbations to better assess robustness and generalization. Within this setup, we systematically explore the design space along three dimensions: 1) foundational components, covering core VLM-policy architectures and action learning objectives; 2) perception essentials: examining the role of visual, language, and proprioceptive inputs; and 3) action modeling perspective: investigating designs and auxiliary objectives that facilitate action generation. We conduct more than 500 distinct experiments over the above three dimensions, and distill 12 key findings that together form a practical recipe for building strong VLA models, summarized in Fig. 2.

We highlight several findings that we believe are novel and noteworthy for the field: 1) a soft connection between the VLM and the policy module performs slightly better than both loose and tight coupling strategies; 2) video inputs, even though the VLM is already pretrained on video understanding, still fails to distill useful information for action learning; 3) conditioning proprioceptive input in the VLM yields better performance than either omitting proprioception or injecting it directly into the policy module; and 4) framing action generation as a time-series forecasting problem and incorporating frequency-domain modeling provides an effective and efficient way to improve action prediction.

The outcome of this study is a simple yet effective VLA model, **VLANeXt**, derived directly from the design principles uncovered in our systematic exploration. Rather than relying on aggressive model scaling or task-specific engineering, VLANeXt achieves state-of-the-art performance on both LIBERO (Liu et al., 2023a) and LIBERO-plus (Fei et al., 2025b) (Fig. 1), and adapts effectively to real-world manipulation tasks. These results show that strong VLA performance can emerge from principled design choices within a unified framework. To support further progress in this direction, we release a **unified and easy-to-use codebase** that standardizes training and evaluation while exposing the key components of the VLA design space. The framework is intentionally lightweight and minimally encapsulated, enabling researchers to reproduce our findings, probe alternative design choices, and build new VLA variants on top of a shared, transparent foundation, which is available at `https://github.com/DravenALG/VLANeXt`.

## 2. Recipes for Building Strong VLA Models

In this section, we detail the step-by-step evolution from a simple baseline to the final VLANeXt model. We organize our exploration along three aspects: foundational compo-

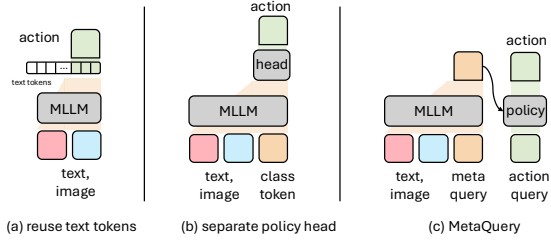

Figure 3. Design choices for the policy module.

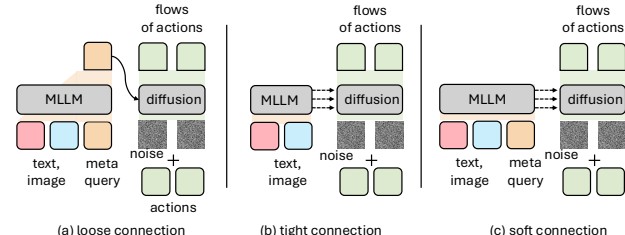

Figure 4. Design choices for the VLM-Policy connection.

nents (Sec. 2.1), perception essentials (Sec. 2.2), and action modeling perspectives (Sec. 2.3). An overview is shown in Fig. 2, with full results in Table 1.

**Evaluation Setup.** We perform the roadmap exploration on LIBERO and LIBERO-plus (Liu et al., 2023a; Fei et al., 2025b). Main experiments are conducted on the spatial suite as our primary testbed, while the resulting insights generalize across the other suites (Object, Goal, and Long), which can also be seen in Table 5.

**Baseline.** Our baseline follows the VLA pipeline introduced in RT-2 (Zitkovich et al., 2023), the origin of VLAs, and later adopted by OpenVLA (Kim et al., 2024a). We use LLaMA as the language backbone (Grattafiori et al., 2024). Since LLaMA does not natively support visual inputs, we also paired our backbone with the SigLIP2 (Tschannen et al., 2025) as the vision encoder, A subset of rarely used text tokens is repurposed as action tokens, enabling action prediction in the same autoregressive framework. Continuous actions are discretized using a simple binning strategy and modeled as classification over bin indices. We intentionally start from this minimal, classical RT2-style setup to provide a clean reference point for analyzing the effects of different design choices. Our implementation adopts a more recent LLaMA version (LLaMA 3.2) but at a smaller scale (3B parameters), compared to OpenVLA (Kim et al., 2024a).

## 2.1. The Foundational Components

In this section, we investigate some core design choices of VLAs, including architectures and training losses.

**Policy Module Design.** Our baseline follows RT-2 (Zitkovich et al., 2023) and OpenVLA (Kim et al., 2024a), reusing text tokens for action classification. We first examine whether an explicit policy head is necessary. To this end, we append a class token to the text and visual embeddings and feed its LLM output into a two-layer policy head (transformer architecture) for action classification (Fig. 3(a)(b)). Results show that introducing a separate policy head performs slightly better than directly reusing text tokens (Table 1), suggesting that *decoupling action prediction from the linguistic token space is beneficial*.

We further investigate whether a more expressive policy module brings additional gains. Specifically, we replace

the single class token with multiple tokens (16) and expand the policy network from 2 to 12 layers, making the design conceptually similar to MetaQuery (Pan et al., 2025) (Fig. 3(c)). This *enlarged policy module yields a significant performance improvement* (Table 1). Our final model adopts this design.

**Action Chunking.** Our baseline predicts actions one step at a time. Here, we evaluate action chunking, which predicts multiple future actions jointly and is known to improve inference efficiency (Kim et al., 2025). Results show that *longer chunk horizons consistently improve action generation performance* (Table 1), suggesting that *modeling a longer temporal window of action provides a more coherent view of the action sequence*. We therefore adopt action chunking with a chunk size of 8.

**Action Learning Objective.** An action chunk is a continuous vector of shape $(t, dim)$. Our baseline discretizes this vector using binning (first normalizing to $-1$ and 1, then dividing into 256 bins) and treats action prediction as classification, following OpenVLA (Kim et al., 2024a). We compare this with alternative objectives, including direct regression (Kim et al., 2025), diffusion-based losses such as DDIM (Song et al., 2021; Zhang et al., 2025c), flow matching (Lipman et al., 2021; Lv et al., 2025), and VQ-VAE–based codebook (codebook size 1024 and each action assigns 3 codes) classification (Van Den Oord et al., 2017; Esser et al., 2021).

Results show that *regression achieves the strongest performance*, *with diffusion-based objectives close behind*, while classification-based approaches perform worst (Table 1). In addition, we also notice that when the performance gets higher, the flow-matching objective will eventually outperform the regression loss, since it can represent precise control signals. We therefore adopt the flow-matching objective. We also observe that classification using the VQ–VAE–based codebook underperforms relative to the binning strategy. We attribute this to the fact that the action spaces are low-rank, meaning a simple binning approach provides sufficient resolution.

**VLM Backbone Capacity.** Our baseline uses LLaMA as the backbone (Grattafiori et al., 2024). We evaluate alternative VLM backbones to study how backbone strength affects VLA performance, including PaliGemma-3B (Beyer

et al., 2024) (used in the $\pi$ series (Black et al., 2024; Intelligence et al., 2025b)) and the Qwen-VL family (Bai et al., 2025a), which represent some of the most capable open-source VLMs currently available.

Results show a consistent trend: *stronger VLM backbones yield better VLA performance* (Table 1), with Qwen3-VL-4B outperforming Qwen3-VL-2B, which in turn outperforms LLaMA-3.2-3B and PaliGemma-3B. We use Qwen3-VL-2B in subsequent experiments as a strong yet efficient choice. This finding differs from (Zhang et al., 2026). A possible reason is that our larger policy module can better exploit the representational capacity of stronger VLMs, whereas the lightweight policy head in (Zhang et al., 2026) may limit such gains. We leave a deeper investigation to future work.

**VLM-Policy Connection.** We next study how different connection strategies between the VLM and the policy module affect performance. Our baseline adopts a MetaQuery-style design (Pan et al., 2025), as discussed in "Policy Module Design". We refer to this design as the loose strategy, where the VLM and policy module are fully decoupled. We compare this with a tight strategy that connects the two modules layer by layer, as in the $\pi$ series (Black et al., 2024; Intelligence et al., 2025b). Inspired by these two designs, we further introduce a soft strategy that also connects them layer by layer but inserts learnable queries as a latent buffer between the modules (Fig. 4). In detail, for the above three connection strategies, we all use the cross attention as the condition technique, and the timestep is conditioned by adaLN, like (Peebles & Xie, 2023).

Results show that the *soft strategy slightly outperforms both loose and tight connections* (Table 1), suggesting that the learnable query buffer helps better transfer useful representations from the VLM's textual space to the policy module's action space. This may be viewed as introducing a latent buffer between the two components, analogous to reasoning in a latent space (Hao et al., 2024). We adopt the soft connection in subsequent models.

## 2.2. The Perception Essentials

In this section, we shift our focus from foundational components to perception, investigating whether and how different modalities (*e.g.*, visual observations and actions) should be provided as inputs to VLAs.

**Temporal Observation History.** We examine whether incorporating temporal observation history improves performance. Our baseline follows OpenVLA (Kim et al., 2024a) and uses only the current frame as input. We extend this to include multiple past frames, leveraging the video capability of the Qwen3-VL-2B backbone (Bai et al., 2025b) for a controlled comparison. Results show that *adding temporal history does not improve action generation* and slightly degrades performance (Table 1), indicating that redundant temporal inputs may introduce noise or distract the model, even though the backbone is already pretrained in video understanding.

**Camera View Horizon.** We study the effect of camera viewpoints on VLA performance. Our baseline uses a single third-person view, following OpenVLA (Kim et al., 2024a). Many robotics datasets (O'Neill et al., 2024; Khazatsky et al., 2024) additionally provide an in-hand wrist camera, allowing choices between single-view and multi-view inputs. Results show that *combining third-person and wrist views significantly improves performance* (Table 1), suggesting that multi-view observations provide complementary geometric cues that help resolve spatial ambiguities.

**Proprioception Conditioning.** We examine the role of proprioception, which provides information about the robot's internal state and motion history. Our baseline, following OpenVLA (Kim et al., 2024a), does not use proprioceptive inputs. We compare three variants: conditioning the VLM, conditioning the policy module, and conditioning both (Fig. 5). In detail, for the VLM part, we will use the proprioception as input, and for the policy part, we will use the action as input to align with the generated action.

Results show that *conditioning proprioception in the VLM yields the best performance* (Table 1). We hypothesize that integrating proprioception at the VLM level allows better fusion with visual and language inputs, whereas injecting it directly into the policy module may reduce reliance of action prediction on visual observations and instructions. Although this appears to differ from the conclusion reported in Zhao *et al.* (Zhao et al., 2025a), where they claim that proprioception is not needed, their study evaluates architectures where proprioception is injected only into the policy module. In that setting, removing proprioception improves performance, which is consistent with our findings.

We further compare three different integration mechanisms, including a linear projector, a transformer-based projector, and a transformer projector with masked reconstruction pretraining (He et al., 2022). The *transformer-based projector performs slightly better* (Table 1); for simplicity, we use the linear projector in the final design.

## 2.3. Action Modelling Perspectives

Here, we examine auxiliary design and training objectives to facilitate action generation.

**World Modelling.** We examine augmenting action prediction with an auxiliary world modeling objective (Lv et al., 2025; Cen et al., 2025b). To maintain relatively fair comparison, we don't use a pretrained visual generator. Instead, we tokenize images using the Emu3.5 image tokenizer (Cui

*Table 1.* Ablation across the VLA design space on LIBERO, LIBERO-plus (spatial suite). Each block varies in one design aspect. LIBERO-plus evaluates robustness under diverse perturbations. Performance improves steadily as effective design choices are incorporated.

| Model | LIBERO | LIBERO-plus | | | | | | | |
|---|---|---|---|---|---|---|---|---|---|
| | Original | Camera | Robot | Language | Light | Background | Noise | Layout | Total |
| ***Foundational Components*** | | | | | | | | | |
| *RT-2 like Baseline* | | | | | | | | | |
| Baseline | 19.8 | - | - | - | - | - | - | - | < 5.0 |
| *Policy Module Design* | | | | | | | | | |
| Baseline | 19.8 | - | - | - | - | - | - | - | < 5.0 |
| Seperate Head | 30.2 | 0.8 | 10.0 | 31.0 | 15.4 | 24.8 | 4.0 | 30.1 | 16.6 |
| Large Policy Module | 64.4 | 0.5 | 12.6 | 79.7 | 34.6 | 32.9 | 8.5 | 63.1 | 34.0 |
| *Action Chunking Horizon* | | | | | | | | | |
| Action Chunk 1 | 64.4 | 0.5 | 12.6 | 79.7 | 34.6 | 32.9 | 8.5 | 63.1 | 34.0 |
| Action Chunk 4 | 75.4 | 5.3 | 28.0 | 67.9 | 42.5 | 50.4 | 14.8 | 70.4 | 40.0 |
| Action Chunk 8 | 74.6 | 5.6 | 26.0 | 85.6 | 56.8 | 55.8 | 11.7 | 63.9 | 43.4 |
| *Action Learning Objective* | | | | | | | | | |
| bin Classification | 74.6 | 5.6 | 26.0 | 85.6 | 56.8 | 55.8 | 11.7 | 63.9 | 43.4 |
| VQ-VAE Classification | 58.8 | 3.2 | 44.3 | 67.2 | 42.5 | 43.8 | 7.1 | 48.3 | 36.5 |
| Regression | 85.4 | 5.1 | 32.3 | 90.5 | 62.7 | 68.6 | 7.7 | 75.6 | 48.4 |
| DDIM | 80.0 | 4.8 | 52.6 | 80.3 | 70.5 | 55.4 | 9.4 | 68.3 | 48.3 |
| Flow Matching | 80.0 | 7.2 | 34.6 | 79.2 | 46.9 | 57.8 | 11.1 | 77.4 | 45.0 |
| *VLM Backbone Capacity* | | | | | | | | | |
| Paligemma | 69.8 | 1.1 | 17.1 | 32.1 | 22.9 | 32.6 | 2.8 | 24.9 | 18.9 |
| LLaMA3.2 + SigLip | 80.0 | 7.2 | 34.6 | 79.2 | 46.9 | 57.8 | 11.1 | 77.4 | 45.0 |
| Qwen3VL-2B | 90.0 | 9.6 | 42.0 | 74.6 | 75.0 | 68.6 | 27.9 | 83.6 | 53.7 |
| Qwen3VL-4B | 95.8 | 12.2 | 66.0 | 93.3 | 89.0 | 81.0 | 29.9 | 88.6 | 64.8 |
| *VLM-Policy Connection* | | | | | | | | | |
| Loose Connection | 90.0 | 9.6 | 42.0 | 74.6 | 75.0 | 68.6 | 27.9 | 83.6 | 53.7 |
| Tight Connection | 90.0 | 14.4 | 51.7 | 81.0 | 68.5 | 67.8 | 25.1 | 82.1 | 55.4 |
| Soft Connection | 91.8 | 11.8 | 58.3 | 89.2 | 72.9 | 74.4 | 19.9 | 72.5 | 56.2 |
| ***Perception Enssentials*** | | | | | | | | | |
| *Temporal Observation History* | | | | | | | | | |
| Current Frame Image | 91.8 | 11.8 | 58.3 | 89.2 | 72.9 | 74.4 | 19.9 | 72.5 | 56.2 |
| Temporal Observation History | 85.0 | 7.2 | 68.6 | 51.5 | 65.8 | 62.0 | 20.8 | 80.8 | 50.2 |
| *Camera View Horizon* | | | | | | | | | |
| Third-person camera view | 91.8 | 11.8 | 58.3 | 89.2 | 72.9 | 74.4 | 19.9 | 72.5 | 56.2 |
| Multiview (third-person + wrist) | 97.6 | 64.9 | 54.0 | 91.8 | 97.7 | 93.0 | 85.5 | 90.1 | 80.5 |
| *Proprioception Conditioning* | | | | | | | | | |
| No Proprioception Input | 97.6 | 64.9 | 54.0 | 91.8 | 97.7 | 93.0 | 85.5 | 90.1 | 80.5 |
| Proprioception to VLM | 98.0 | 87.2 | 62.2 | 86.2 | 98.3 | 93.8 | 92.0 | 96.6 | 87.7 |
| Proprioception to Policy | 96.2 | 62.8 | 69.1 | 92.3 | 92.5 | 96.5 | 87.2 | 88.3 | 83.4 |
| Proprioception to VLM & Policy | 97.6 | 77.9 | 73.4 | 82.3 | 90.1 | 94.6 | 90.6 | 88.8 | 84.8 |
| Linear Projector | 98.0 | 87.2 | 62.2 | 86.2 | 98.3 | 93.8 | 92.0 | 96.6 | 87.7 |
| Transformer Projector | 96.4 | 96.8 | 58.0 | 84.6 | 95.2 | 98.8 | 96.9 | 93.3 | 88.8 |
| Transformer Projector & MAE | 97.0 | 91.1 | 51.1 | 89.9 | 72.8 | 86.9 | 78.7 | 85.6 | 78.9 |
| ***Action Modelling Perspectives*** | | | | | | | | | |
| *World Modelling Perspective* | | | | | | | | | |
| Normal | 98.0 | 87.2 | 62.2 | 86.2 | 98.3 | 93.8 | 92.0 | 96.6 | 87.7 |
| World Modelling | 98.0 | 94.4 | 80.3 | 76.9 | 99.7 | 98.8 | 93.2 | 93.2 | 90.3 |
| *Time Series Forecasting Perspective* | | | | | | | | | |
| Normal | 98.0 | 87.2 | 62.2 | 86.2 | 98.3 | 93.8 | 92.0 | 96.6 | 87.7 |
| Frequency Domain Loss | 99.0 | 95.7 | 78.6 | 86.9 | 99.7 | 98.8 | 98.0 | 96.6 | 93.1 |

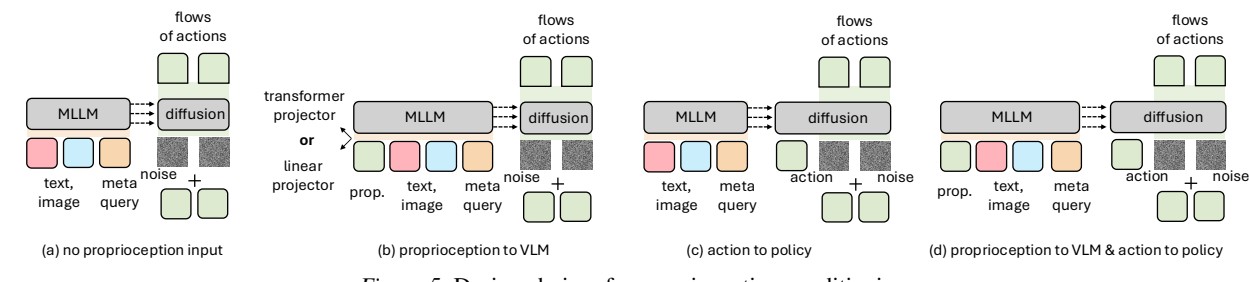

Figure 5. Design choices for proprioception conditioning.

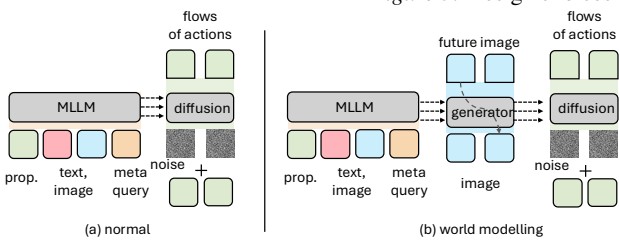

Figure 6. Augmenting action prediction with an auxiliary world modeling objective.

et al., 2025) and predict future image tokens with a next-token objective. The target is the future frame at a fixed horizon (8 steps, aligned with the action chunk length). The visual generation module is inserted between the VLM and the policy module with layer-wise connections (Fig. 6). *Adding world modeling improves action generation performance* (Table 1), indicating that predicting future observations is beneficial. However, it nearly triples training time, substantially increasing computational cost. We therefore exclude world modeling from the final recipe.

**Time Series Forecasting.** We also explore facilitating action generation from a time-series forecasting perspective. Inspired by frequency-domain modeling in time-series prediction (Zhou et al., 2022; Yi et al., 2023; Yang et al., 2024; Wang et al., 2025b), we introduce a simple auxiliary loss that minimizes the MSE between predicted and ground-truth actions in the frequency domain. We use the discrete cosine transform (Ahmed et al., 1974) to convert the action to the frequency domain, and assign higher weights to low-frequency components and lower weights to high-frequency components, as high-frequency components are often noisier.

This strategy *improves action generation performance*, slightly surpassing the world modeling objective while adding negligible training overhead (Table 1). The gain likely arises because it serves as a regularization term to avoid the model overfitting to the jitter in trajectory, which mainly improves the model's generalization ability.

### 2.4. Summary of Recipes

Starting from a classical RT-2/OpenVLA-style baseline, we find that strong VLA performance emerges from a series of principled design choices. Beneficial changes include: replacing token reuse with a deeper, dedicated policy module;

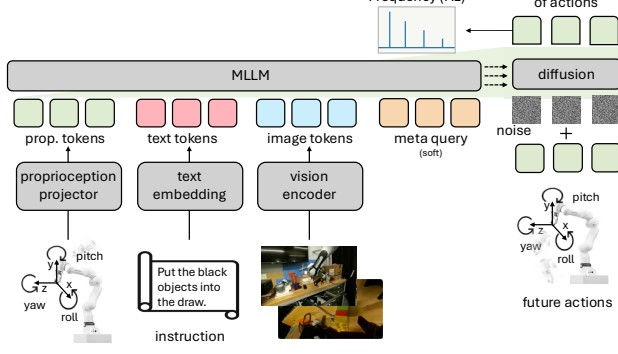

Figure 7. **VLANeXt architecture**. Multi-view visual inputs, language instructions, and proprioception are tokenized and processed by a multimodal LLM, with meta queries enabling soft interaction with the policy module. Action chunks are predicted using flow matching and further regularized by a frequency-domain objective.

adopting action chunking to model longer temporal action horizons; using continuous objectives such as flow matching (with regression also effective under simple distributions); employing a stronger VLM backbone (Qwen3-VL-2B as an effective–efficient choice); and connecting the VLM and policy module through soft, layer-wise interactions with learnable query buffers.

On the perception side, multi-view inputs (third-person + wrist) and VLM-side proprioception conditioning improve performance, while redundant temporal observation history is unnecessary. Moreover, adding a lightweight frequency-domain auxiliary loss further boosts action generation with negligible cost. Although world modeling also improves performance, its substantially higher training cost makes it less practical. Together, these choices form a practical recipe for building a strong and efficient VLA model, which we call VLANeXt.

## 3. Benchmarks Evaluations

### 3.1. Settings

To evaluate both standard performance and generalization robustness, we employ the LIBERO ecosystem. We first evaluate our VLANeXt on the standard LIBERO benchmark (Liu et al., 2023a), which provides four distinct categories (Spatial, Object, Goal, and Long) to test the task learning ability, each providing 500 expert demonstrations

across 10 tasks to assess policy generalization to different spatial layouts, objects, goals, and long-horizon tasks.

To test the generalization boundaries of our model further, we evaluate our method on LIBERO-plus (Fei et al., 2025b). Unlike the static conditions in standard LIBERO, LIBERO-plus introduces systematic variations to the evaluation episodes, comprising 10,030 demonstrations across the above four suites in LIBERO, with perturbations in visual (*e.g.*, lighting, background, camera pose), physical (*e.g.*, object layout, robot state), and semantic (*e.g.*, language instruction rewrites) dimensions.

Following the standard setting in OpenVLA (Kim et al., 2024a), we train our models on the modified LIBERO dataset for each suite (Spatial, Object, Goal, and Long), and evaluate performance on both the LIBERO and LIBERO-plus benchmarks (which include unseen perturbations) for the corresponding suite. For fair comparisons across different design choices, we directly fine-tune all models on the LIBERO dataset. All experiments in our recipes use 10,000 training steps with a batch size of 256. The learning rate is set to $1 \times 10^{-4}$ for models smaller than 3B parameters and $5 \times 10^{-5}$ otherwise in our explorations.

### 3.2. LIBERO Benchmark Results

On the LIBERO benchmark, we compare our method against two categories of approaches: (i) direct policy learning methods that are trained solely on robotic datasets, and (ii) VLA methods that leverage knowledge from pretrained VLMs for policy learning. For direct policy learning methods, we include Diffusion Policy (Chi et al., 2025), Octo (Ghosh et al., 2024), and MDT (Reuss et al., 2024). For VLA methods, we compare against OpenVLA (Kim et al., 2024a), TraceVLA (Zheng et al., 2025a), SpatialVLA (Qu et al., 2025), WorldVLA (Cen et al., 2025b), CoT-VLA (Zhao et al., 2025b), $\pi_0$ (Black et al., 2024), $\pi_0$-Fast (Pertsch et al., 2025), NORA (Hung et al., 2025), SmolVLA (Shukor et al., 2025), UniVLA (Wang et al., 2025e), FLOWER (Reuss et al., 2025), and OpenVLA-OFT (Kim et al., 2025).

The comparison results are shown in Table 2. As we can observe, following our recipes allows us to build a strong VLA that achieves state-of-the-art performance, demonstrating the effectiveness of the design choices.

### 3.3. LIBERO-plus Benchmark Results

For the LIBERO-plus benchmark, we compare our model with several VLA models such as OpenVLA (Kim et al., 2024a), WorldVLA (Cen et al., 2025b), NORA (Hung et al., 2025), UniVLA (Wang et al., 2025e), $\pi_o$ (Black et al., 2024), $\pi_o$-Fast (Pertsch et al., 2025), and OpenVLA-OFT (Kim et al., 2025).

*Table 2.* LIBERO benchmark performance. The results are shown in success rate (%). S, O, G, L: Spatial, Object, Goal, and Long suites, respectively. We color the best and second best results.

| Model | S | O | G | L | Avg |
|---|---|---|---|---|---|
| *Baseline Direct Policy Models* | | | | | |
| Diffusion Policy | 78.3 | 92.5 | 68.3 | 50.5 | 72.4 |
| Octo | 78.9 | 85.7 | 84.6 | 51.5 | 75.1 |
| MDT | 78.5 | 87.5 | 73.5 | 64.8 | 76.4 |
| *Baseline VLA Models* | | | | | |
| TraceVLA | 84.6 | 85.2 | 75.1 | 54.1 | 74.8 |
| OpenVLA | 84.7 | 88.4 | 79.2 | 53.7 | 76.5 |
| SpatialVLA | 88.2 | 89.9 | 78.6 | 55.5 | 78.1 |
| WorldVLA | 85.6 | 89.0 | 82.6 | 59.0 | 79.1 |
| CoT-VLA | 87.5 | 91.6 | 87.6 | 69.0 | 83.9 |
| $\pi_0$-Fast | 96.4 | 96.8 | 88.6 | 60.2 | 85.5 |
| $\pi_0$ | 90.0 | 86.0 | 95.0 | 73.0 | 86.0 |
| NORA | 92.2 | 95.4 | 89.4 | 74.6 | 87.9 |
| SmolVLA | 93.0 | 94.0 | 91.0 | 77.0 | 88.8 |
| UniVLA | 96.5 | 96.8 | 95.6 | 92.0 | 95.2 |
| FLOWER | 97.5 | 99.1 | 96.1 | 94.9 | 96.9 |
| $\pi_{0.5}$ | 98.8 | 98.2 | 98 | 92.4 | 96.9 |
| OpenVLA-OFT | 97.6 | 98.4 | 97.9 | 94.5 | 97.1 |
| *Ours* | | | | | |
| **VLANeXt** | **99.0** | **99.2** | 96.6 | 94.8 | **97.4** |

As shown in Table 3, the proposed VLANeXt model demonstrates strong generalization ability across different types of unseen perturbations. Moreover, our model shows a significant improvement (13% in success rate) over the state-of-the-art method OpenVLA-OFT (Kim et al., 2025) on the LIBERO-plus benchmark compared to previous methods, suggesting the effectiveness of the explored recipes.

## 4. Real-World Evaluations

To comprehensively assess the performance of our method, we additionally evaluate it in real-world deployments.

### 4.1. Settings

We design four tasks, including two single-arm tasks and two bimanual tasks, to evaluate our method. The single-arm tasks include table cleaning, which involves picking up objects from a table and placing them into a container, and drawer manipulation, where the robot opens a drawer, places objects inside, and closes it. The bimanual tasks include basket lifting, which requires lifting a basket using both hands, and bimanual table cleaning, where two arms coordinate to collect objects from a table and place them into a container. The single-arm experiments use Franka Emika, while the bimanual experiments are conducted on the Aloha system (Zhao et al., 2023). A visualization of the experimental setup for each task is depicted in Figure 8.

For training, we collect 50 episodes per task and evaluate each model over 20 trials, reporting the success rate. We

*Table 3.* LIBERO-plus benchmark performance. The results are shown in success rate (%). We color the **best** and second best results (in average). The complete per-suite results of the listed methods can also be found in (Fei et al., 2025b).

| Model | Suite | Camera | Robot | Language | Light | Background | Noise | Layout | Total |
|---|---|---|---|---|---|---|---|---|---|
| *Baseline VLA Models* | | | | | | | | | |
| OpenVLA | Average | 0.8 | 3.5 | 23.0 | 8.1 | 34.8 | 15.2 | 28.5 | 15.6 |
| WorldVLA | Average | 0.1 | 27.9 | 41.6 | 43.7 | 17.1 | 10.9 | 38.0 | 25.0 |
| NORA | Average | 2.2 | 37.0 | 65.1 | 45.7 | 58.6 | 12.8 | 62.1 | 39.0 |
| UniVLA | Average | 1.8 | 46.2 | 69.6 | 69.0 | 81.0 | 21.2 | 31.9 | 42.9 |
| $\pi_0$ | Average | 13.8 | 6.0 | 58.8 | 85.0 | 81.4 | 79.0 | 68.9 | 53.6 |
| $\pi_0$-Fast | Average | 65.1 | 21.6 | 61.0 | 73.2 | 73.2 | 74.4 | 68.8 | 61.6 |
| | Spatial | 88.3 | 40.0 | 80.5 | 98.3 | 97.3 | 96.3 | 93.9 | 84.0 |
| | Object | 38.9 | 25.4 | 99.0 | 73.7 | 97.6 | 72.3 | 71.8 | 66.5 |
| OpenVLA-OFT | Goal | 62.0 | 25.2 | 53.2 | 93.9 | 92.5 | 75.2 | 59.1 | 63.0 |
| | Long | 38.7 | 38.2 | 87.0 | 89.4 | 86.8 | 63.5 | 76.9 | 66.4 |
| | Average | 56.4 | 31.9 | 79.5 | 88.7 | **93.3** | 75.8 | 74.2 | 69.6 |
| *Ours* | | | | | | | | | |
| | Spatial | 95.7 | 78.6 | 86.9 | 99.7 | 98.8 | 98.0 | 96.6 | 93.1 |
| | Object | 99.5 | 48.5 | 98.6 | 99.3 | 84.7 | 99.8 | 78.2 | 86.5 |
| VLANeXt | Goal | 96.6 | 63.6 | 51.5 | 97.5 | 70.8 | 96.8 | 63.9 | 76.2 |
| | Long | 69.7 | 72.0 | 90.3 | 86.9 | 75.8 | 81.7 | 84.3 | 79.7 |
| | Average | **90.4** | **65.7** | **81.8** | **95.9** | 82.5 | **94.1** | **80.8** | **83.9** |

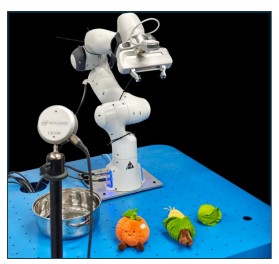
(a) Clean Table

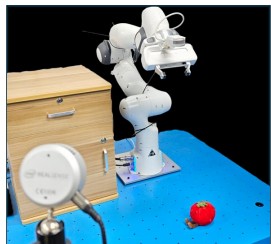
(b) Open Drawer and Place Object

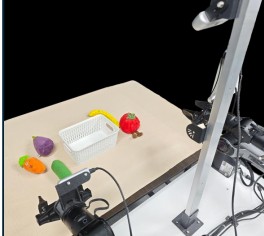
(c) Lifting

(d) Bimanual Clean Table

*Figure 8.* Our single-arm and bimanual arm tasks for the real-world experiments.

first pretrain the model on the DROID dataset (Khazatsky et al., 2024) for 100k steps, then fine-tune it on each task for 20k steps with a learning rate of $1 \times 10^{-4}$. Because DROID contains only single-arm data, adapting the model to bimanual tasks requires reinitializing the proprioception projector and the final layer of the action generation module, while keeping all other pretrained weights.

### 4.2. Results

We compare against two representative VLA baselines, OpenVLA-OFT (Kim et al., 2025) and $\pi_0$ (Black et al., 2024). We load their pretrained checkpoints and fine-tune

*Table 4.* Real-world evaluation results. Results are shown with (success count/total count). We color the **best** and second best .

| Model | Single Arm | | Bimanual Arm | |
|---|---|---|---|---|
| | Clean | Drawer | Clean | Lifting |
| *Baseline VLA Models* | | | | |
| OpenVLA-OFT | 7/20 | 7/20 | 5/20 | 9/20 |
| $\pi_0$ | 10/20 | 8/20 | 10/20 | 13/20 |
| *Ours* | | | | |
| **VLANeXt** | **14/20** | **11/20** | **11/20** | **15/20** |

them on each task in the same manner as our method to ensure a fair comparison. The results are shown in Table 4. As can be seen, our model performs well in real-world experiments, demonstrating that the recipes we propose lead to a strong VLA model that can be effectively deployed in real-world settings. In addition, even without bimanual training, our method can adapt to bimanual robotics tasks with decent performance, demonstrating the cross-embodiment adaptability of the method. Additional video demonstrations of our experimental results are provided in the supplementary materials.

## 5. Conclusion

This work moves toward a more systematic understanding of VLA models. Rather than introducing another standalone architecture, we revisit the VLA pipeline and show that many gains arise from principled design choices within a unified framework. In particular, how the VLM interacts with the policy module, how multimodal signals such as proprioception are fused, and how temporal structure in actions is modeled all play central roles. Several observations carry broader implications. Modest architectural refine-

ments, such as soft VLM–policy coupling or VLM-side proprioception conditioning, can meaningfully influence performance, indicating that where information is injected matters as much as what information is used. Viewing action generation as structured sequence modeling, for example, through frequency-domain objectives, also shows that ideas from time-series learning transfer effectively to robotics. Meanwhile, richer objectives like world modeling improve performance but introduce notable computational overhead, highlighting the importance of efficiency-aware design.

We hope this work encourages a shift from ad-hoc model variations toward more controlled exploration of the VLA design space. By releasing a unified, lightweight framework, we aim to support systematic studies and shared progress. Extending this perspective to more diverse embodiments, longer-horizon reasoning, extensive mid-training, and richer world-interaction objectives remains an important direction for future research.

## Acknowledgements

This research is supported by cash and in-kind funding from NTU S-Lab and industry partner(s). It is also supported by Singapore MOE AcRF Tier 2 (MOE-T2EP20224-0003), and by Guangdong Key Research and Development Program (No.2024B0101040004, No. 2025B0909020002). Also, thanks to victkk (Zicheng Zhang) for giving us valuable suggestions to improve the paper.

## Impact Statement

This paper presents work aimed at advancing the field of machine learning, specifically Vision-Language-Action models for robotic control. While our work contributes to the development of more capable embodied agents, we believe that its potential societal implications fall within well-established discussions in the field and therefore do not require special emphasis here.

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

# A. More Experimental Results

## A.1. Qualitative Experiments

We present more demos of our model on the LIBERO and LIBERO-plus benchmarks, as well as in real-world settings (see Figures 10, 11, and 9). Video demonstrations of our experimental results are provided in the page.

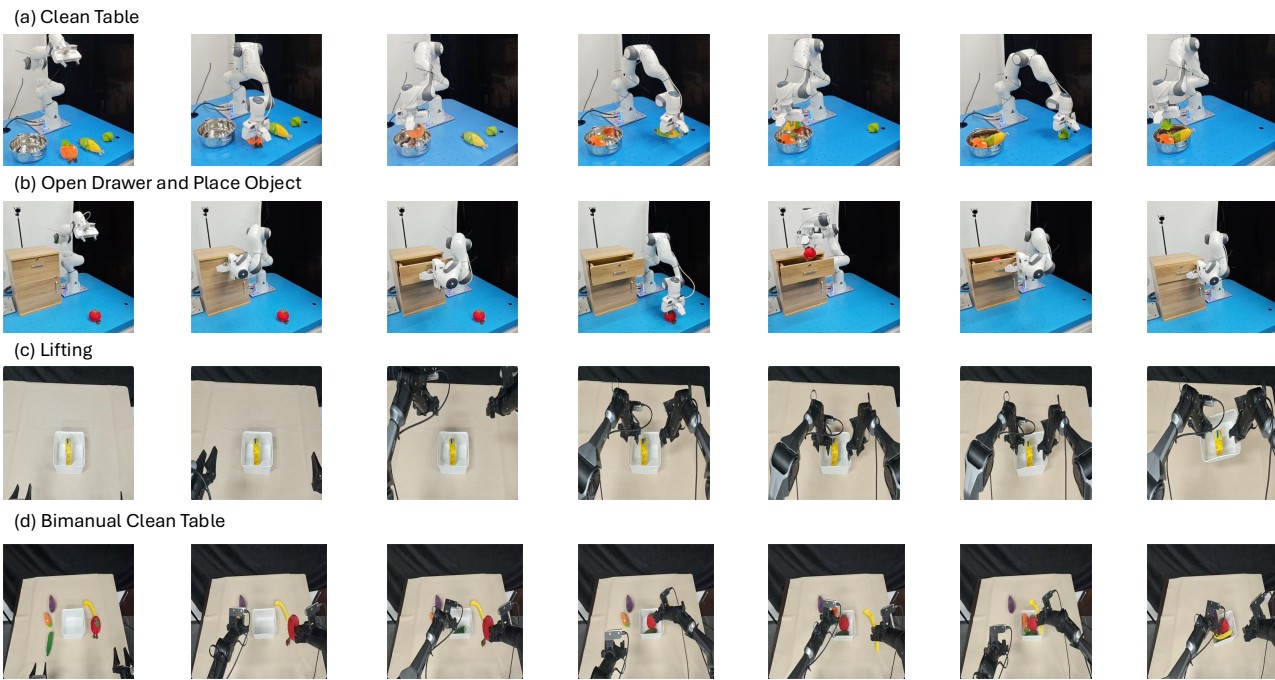

*Figure 9.* Qualitative experiments of our method in real-world tasks.

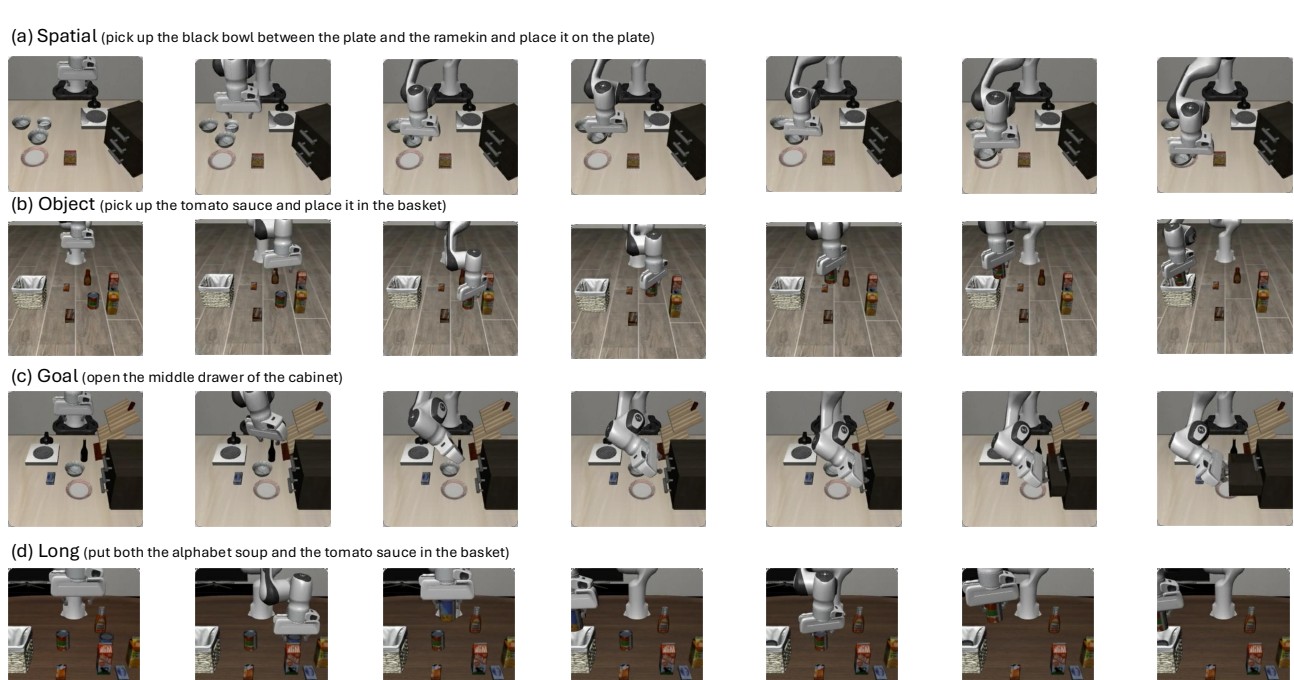

*Figure 10.* Qualitative experiments of our method in the four suites of the LIBERO benchmark.

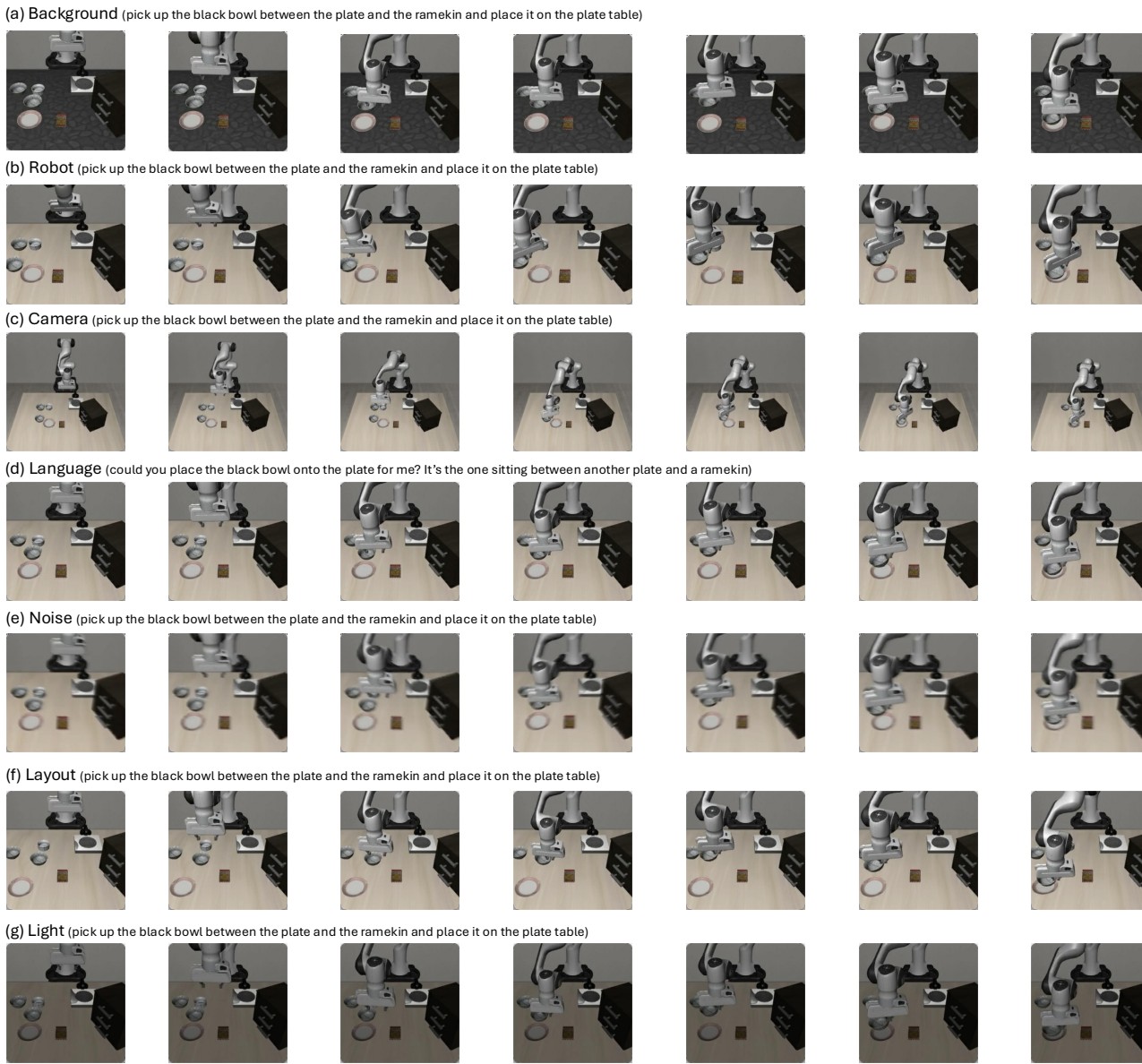

*Figure 11.* Qualitative experiments of our method in the 7 types of perturbations in the same task in the LIBERO-plus benchmark.

## A.2. More Ablation Experiments

We present more ablations of our model on the other suites (Object, Goal, and Long) on the LIBERO and LIBERO-plus benchmarks, as well as in the clean table task in real-world settings. The ablations in other suites on the LIBERO and LIBERO-plus benchmark follow the same experimental setup of our main experiments in Table 1, and the clean table task experiment also follows the real-world setup in Table 4 in the main paper. It can be seen from Table 5 and Table 6 that the results are consistent with the experiments in the Spatial suite on LIBERO and LIBERO-plus benchmarks (Table 1), which shows the robustness of our recipe.

## A.3. Detailed Experimental Settings

We present the detailed training configuration of our final model. The same parameters are used across all four suites, which also shows the robustness of our recipe.

*Table 5.* Results on LIBERO and LIBERO-Plus Benchmarks. We color the best and second best results. Freq. means the frequency-domain loss from the time-series forecasting perspective. Prop. means the input of proprioception to the VLM. Mv. means multi-view inputs. And soft means soft connection.

| Model | LIBERO | | | | | LIBERO-Plus | | | | |
|---|---|---|---|---|---|---|---|---|---|---|
| | Spatial | Object | Goal | Long | Avg | Spatial | Object | Goal | Long | Avg |
| VLANeXt − freq. − prop. − mv. − soft | 90.0 | 31.2 | 93.6 | 83.4 | 74.6 | 53.7 | 38.2 | 36.7 | 50.9 | 44.9 |
| VLANeXt − freq. − prop. − mv. | 91.8 | 61.8 | 94.8 | 90.2 | 84.7 | 56.2 | 41.4 | 47.3 | 54.0 | 49.7 |
| VLANeXt − freq. − prop. | 97.6 | 98.8 | 96.0 | 93.8 | 96.6 | 80.5 | 74.4 | 69.0 | 68.1 | 73.0 |
| VLANeXt − freq. | 98.0 | **99.4** | 95.6 | **95.0** | 97.0 | 87.7 | 77.9 | 71.3 | 71.4 | 77.1 |
| VLANeXt | **99.0** | 99.2 | **96.6** | 94.8 | **97.4** | **93.1** | **86.5** | **76.2** | **79.7** | **83.9** |

*Table 6.* Ablation Study on Real-World Clean Table. We color the best.

| Model | Real-world clean table |
|---|---|
| VLANeXt w/o. action chunk | 1/10 |
| VLANeXt w/o. multiview | 6/10 |
| VLANeXt w/o. soft connection | 5/10 |
| VLANeXt w/o. proprioception | 4/10 |
| VLANeXt | **7/10** |

*Table 7.* Hyperparameters for VLANeXt training on LIBERO and LIBERO-plus benchmark, four suites.

| Hyperparameter | Value |
|---|---|
| *Optimization* | |
| Optimizer | AdamW |
| Learning Rate | $1.0 \times 10^{-4}$ |
| Batch Size | 256 |
| Training Steps | 10,000 |
| Warmup Steps | 500 |
| Lr Scheduler | Cosine Decay |
| Weight Decay | 0.01 |
| Max Grad Norm | 1.0 |
| Backbone Update | Full Finetune |
| *Data & Augmentation* | |
| Observation Modality | Image |
| Camerate Video | Multi-view |
| Proprioception to VLM | true |
| Proprioception to Policy | false |
| Transformer Proprioception Projector | false |
| History Proprioception Size | 8 |
| Action Chunk Size | 8 |
| Augmentation Type | Random Crop, Color Jitter |
| Crop Scale / Ratio | [0.8, 1.0] / [0.9, 1.1] |
| Color Jitter (B/C/S/H) | $0.2/ \pm 0.2/ \pm 0.2/0.05$ |
| *Architecture* | |
| VLM Backbone | Qwen3-VL-2B-Instruct |
| Condition Type | Soft Connection |
| Num of Queries | 16 |
| Action Loss | Diffusion Loss |
| Schedule | Flow Matching |
| Diffusion Hidden Dim | 1024 |
| Diffusion Depth | 29 |
| Diffusion Heads | 16 |
| Frequency Domain Loss Weight | 0.5 |

## B. Revisiting Robot Learning and VLA Models

Robot learning aims to apply machine learning techniques to robotic control, empowering robots to interact with the physical world and acquire diverse skills (Ravichandar et al., 2020). In the context of robot learning, tasks are generally categorized into locomotion and manipulation based on the active components being controlled. Locomotion is designed to maintain the stability and balance of the robot base, enabling mobility for legged systems such as quadrupeds or humanoids (Peng et al., 2018; Kumar et al., 2021; Lee et al., 2020; Margolis & Agrawal, 2023; Kim et al., 2024b; Liu et al., 2025). Since these tasks typically possess explicit objectives, they frequently use Reinforcement Learning (RL). In contrast, robotic manipulation focuses on controlling the robotic arm or whole body to execute a diverse array of interactive tasks. Previously, manipulation was usually focused on specific tasks, such as gripper grasping (Fang et al., 2020; Wang et al., 2021; Wu et al., 2024b; Cai et al., 2024; Wang et al., 2025a), dexterous hand grasping (Liu et al., 2023c; Li et al., 2023a; Huang et al., 2023a; Xu et al., 2024; Wei et al., 2024; 2025), dynamic grasping (Liu et al., 2023b; Chen et al., 2024), and non-prehensive manipulation (Mason, 1999; Mordatch et al., 2012; Lyu et al., 2025). These tasks can typically be formulated with well-defined task priors, making them relatively easier to learn and generalize. More recently, general-purpose manipulation has gradually become a central research direction (Brohan et al., 2023; Chi et al., 2025; Wang et al., 2024; Liu et al., 2024; Ghosh et al., 2024). However, due to the diversity of task goals and manipulated objects, it is often difficult to design explicit reward functions or leverage well-defined task priors for such settings. As a result, imitation learning (IL) has been widely adopted, enabling robots to acquire complex manipulation skills directly from expert demonstrations. In this paper, we primarily focus on general-purpose robotic manipulation through imitation learning. Robotic manipulation methods can be divided into standard action policies (Huang et al., 2022; Shridhar et al., 2023; Radosavovic et al., 2023; Huang et al., 2023b; Ze et al., 2024; Lu et al., 2024; Huang et al., 2025c; Jiang et al., 2025; Zheng et al., 2025b; Kuang et al., 2025; Zhou et al., 2025a; Zhao et al., 2025a) and video action policies (Janner et al., 2022; Wu et al., 2023; 2024a; Du et al., 2023; Cheang et al., 2024; Tian et al., 2024; Li et al., 2025c). The former framework naively inputs instructions and visuals, outputting the actions to complete the tasks, while the latter pipeline predicts future videos together with action generation, with the claim that these world modeling abilities can help understand the task better and generate the actions more accurately.

In recent years, with the triumph of large foundation models, integrating these tremendous models into robot learning, specifically called Vision-Language-Action (VLA) Models, has become a prominent trend (Ma et al., 2024). This paradigm was pioneered by the groundbreaking RT-2 (Zitkovich et al., 2023), which formally introduced the concept of VLA. Subsequently, researchers across academia and industry have developed a diverse array of VLA models (O'Neill et al., 2024; Li et al., 2023b; 2024; Kim et al., 2024a; Black et al., 2024; Team et al., 2025; Hung et al., 2025; Kim et al., 2025; Shukor et al., 2025; Intelligence et al., 2025b;a; Liu et al., 2026). These newer iterations address specific challenges, such as leveraging 3D spatial information (Zhen et al., 2024; Bhat et al., 2025; Zhang et al., 2025a; Qu et al., 2025), exploiting intermediate data (like subtasks decomposition, future frame prediction or robot trajectory traces prediction) (Zheng et al., 2025a; Zhao et al., 2025b; Zhang et al., 2025c; Lv et al., 2025; Zhong et al., 2025; Liang et al., 2025; Lee et al., 2025; Cen et al., 2025b;a; Wang et al., 2025e; Zhang et al., 2025b; Song et al., 2025) to enhance action generation, and designing post-training optimization like planning or reinforcement learning to adapt to specific environment (Guo et al., 2025; Zhang et al., 2025d; Bai et al., 2025c; Tan et al., 2025; Li et al., 2025b; Fei et al., 2025a; Chen et al., 2025b; Huang et al., 2025a; Lu et al., 2025; Xiao et al., 2025a;b). Additionally, a subset of VLAs explores some niche but important aspects (Zhou et al., 2025b; Wang et al., 2025c; Kareer et al., 2025; Shi et al., 2025; Fu et al., 2025; Pertsch et al., 2025; Goyal et al., 2025; Zhang et al., 2026; Zhou et al., 2026), such as latent actions (Ye et al., 2024; Bi et al., 2025), lightweight VLAs (Wen et al., 2025; Li et al., 2025a; Reuss et al., 2025; Wang et al., 2025d) and VLAs in specific domains (Chen et al., 2025a; Bjorck et al., 2025; Ding et al., 2025; Huang et al., 2025b). Despite their different emphases, most VLA models follow a similar pipeline that builds on pretrained LLMs or VLMs to process visual observations and language instructions and produce action-relevant representations for policy learning, yet this pipeline admits many design choices spanning model interfacing, policy training, perception, and action modeling. As a result, early VLA research remains a "primordial soup": rich in ideas but insufficiently structured, and the diversity of existing frameworks, together with inconsistent training and evaluation protocols, makes it difficult to identify truly impactful choices. Toward this end, this work aims to provide a more systematic understanding of this fragmented design space by comprehensively reexamining VLA design spaces under a unified framework and evaluation protocol. We conduct more than 500 distinct experiments over the above three dimensions, and distill 12 key findings that together form a practical recipe for building strong VLA models. The outcome of this study is a simple yet effective VLA model, **VLANeXt**, which achieves state-of-the-art performance on both LIBERO (Liu et al., 2023a) and LIBERO-plus (Fei et al., 2025b) (Fig. 1), and adapts effectively to real-world manipulation tasks.

