# OpenReview forum: "VLANeXt: Recipes for Building Strong VLA Models"
_ICML.cc/2026/Conference — ICML 2026 regular_

### Official Review · Reviewer_Yhj1 · 2026-03-07

**Soundness:** 3
**Presentation:** 3
**Significance:** 3
**Originality:** 2
**Overall Recommendation:** 5
**Confidence:** 4

**Summary:**

This paper presents a systematic study of design choices in VLA models and introduces the VLANeXt based on these findings.  The authors revisit the VLA pipeline and show that many performance gains can be achieved through principled design decisions within a unified framework. In particular, the paper analyzes VLA design from three perspectives: foundational components, perception essentials, and action modeling. These analyses provide useful insights and guidance for future VLA model design. The effectiveness of the proposed design choices is validated through both simulation and real-world experiments.

**Compliance With Llm Reviewing Policy:**

Affirmed.

**Final Justification:**

I appreciate the authors’ detailed response and additional results. The clarification regarding the choice of flow matching and the corrected ablation table have addressed my previous concerns. The explanation of the soft VLM-policy connection is also clearer now. It is nice to have more details of this connection in the final manuscript. With these points clarified, and considering the authors’ responses to other reviewers, I believe the current version is self-contained. While minor flaws may still remain, I will raise my score accordingly.

**Key Questions For Authors:**

See weakness.

**Limitations:**

The paper does not clearly discuss the limitations of VLANeXt. However, since the primary goal of the paper is to systematically evaluate model design choices, a detailed limitation analysis may not be strictly necessary. Still, briefly discussing potential limitations could further strengthen the paper.

**Strengths And Weaknesses:**

**Strengths**
1. The motivation for systematically analyzing VLA design choices is well justified. With the rapid emergence of VLA models, there is still limited consensus on effective architectural and training principles. This paper provides useful guidance for understanding and designing VLA models.
2. The paper is comprehensive and technically solid. Most of the writing and figures are clear and easy to follow.
3. The experiments include both simulation and real-world evaluations, and cover different task configurations such as single-arm and bimanual manipulation, providing a relatively comprehensive empirical validation, although some tasks appear to be relatively simple.
4. The authors mention that they plan to release the code, which will be valuable for the research community and facilitate future studies.
5. I particularly appreciate the idea of $\textit{Time Series Forecasting}$ proposed in the paper, which provides an interesting perspective on modeling action trajectories.

**Weakness**
1. In the $\textit{Action Learning Objective}$ section,
    > Results show that regression achieves the strongest performance, with diffusion-based objectives close behind, while
classification-based approaches perform worst (Table 1)...

    The motivation for choosing flow matching is not entirely clear. According to Table 1, regression achieves the strongest performance, while flow matching performs worse. However, the paper ultimately adopts flow matching without sufficiently explaining why regression is not used instead. This design decision would benefit from further clarification.


2. I find the ablation results in Table 1 somewhat confusing. In the Temporal Observation History block, “Current Frame Image” achieves a score of 91.8 on the LIBERO benchmark and is selected as the better configuration.  In the subsequent “Third-person camera view” row, the LIBERO result remains the same, which appears reasonable since the authors adopt the current frame setting. However, the result on the LIBERO-plus benchmark differs. Since the ablation table seems to follow a sequential design where each block builds upon the previous best configuration, it is unclear why these values do not match. Clarifying whether these experiments were conducted with different settings or baselines would help improve the readability and interpretability of the table.

3. In the $\textit{VLM-Policy Connection.}$, the implementation of the proposed soft connection between the VLM and the policy module is not fully explained. The paper mainly refers to Figure 4 but does not provide sufficient textual description or equations to clearly illustrate how this mechanism works. As a result, it is difficult to fully understand the design.

4. In Figure 5, the input corresponding to proprioception is labeled as “actions,” which may be misleading. In robotics, actions typically refer to the outputs of the policy rather than the robot state. If this input actually represents proprioceptive observations (e.g., joint states or current end-effector poses), clarifying the terminology would help avoid potential confusion.

If the authors could clarify these points and address my misunderstandings, I would be happy to reconsider my score.

---

> ### Author Rebuttal · Authors · 2026-03-31
>
> **Q1: Choice in "Action Learning Objective" section**
>
> **A1:** We choose flow matching in the "Action Learning Objective" section because we observe that when the baseline performance gets better, flow matching **performs slightly better** than regression, which can represent precise control signals, thus we choose flow matching. The performance can be seen below.
>
>
> | Model | LIBERO-Spatial | LIBERO-plus-Spatial |
> | --- | --- | --- |
> | Moderate baseline w. regression (results reported in the main paper) | 85.4 | 48.4 |
> | Moderate baseline w. flow matching (results reported in the main paper)  | 80.0 | 45.0 |
> | VLANeXt w. regression | 97.2 | 92.4 |
> | VLANeXt | 99.0 | 92.8 |
>
>
> Our work not only explores a straight line. Behind, we try **more than 400 distinct combinations** in different design spaces and setups, and the roadmap is just a distillation from these explorations. Thus, we will choose the design that gets the best performance at the end for each step.
>
> Thank you for the careful review. We will give an explanation in our revised version to explain why we choose flow matching.
>
> ---
>
> **Q2: Data misalignment in the ablation table**
>
> **A2:** The misaligned data is a typo; we copied the wrong data. We will revise it in our revised version. Thank you for the careful review.
>
> The result with average performance 57.2 is obtained with the same design protocol but larger policy network. The result with average performance 56.2 is the one aligned with the whole table's setup.
>
> ---
>
> **Q3: Explain the soft VLM-policy connection**
>
> **A3:** The soft VLM-policy connection is an enhanced version of the tight connection.
>
> Tight connection connects the VLM with the policy layer-by-layer, leveraging the layer-wise features of the LLM as the condition for the policy network. The VLM features are conditioned by cross attention, and the timestep is conditioned by adaLN, like DiT[1].
>
> Then, considering that the layer-wise features in the VLM are for textual output, whose space is quite different from the action output. Thus, we further add a group of learnable queries into the VLM, before the layer-by-layer connection, which serve as the buffer to transfer from text space to action space.
>
> The formula is as follows:
> $$
> f^{(i)}, metaquery^{(i)} = VLM^{(i)}(f^{(i-1)}, metaquery^{(i-1)});
> $$
> $$
> h^{(i)} = policy^{(i)}(h^{(i-1)}, f^{(i)}, metaquery^{(i)}, t) = CrossAttention(h^{(i-1)}, f^{(i)}, metaquery^{(i)}) * scaler(t) + shift(t)
> $$
> where $i$ stands for layer, and $t$ is the timestep in flow matching.
>
> We will explain this in more detail in our revised version.
>
> ---
>
> **Q4: Use "Action" to represent "Proprioception"**
>
> **A4:** Thank you for your constructive feedback. We will carefully distinguish the "action" and the "proprioception" terms in our revised version.
>
> [1] Peebles W, Xie S. Scalable diffusion models with transformers. ICCV. 2023: 4195-4205.

---

> > ### Author Rebuttal · Reviewer_Yhj1 · 2026-04-02
> >
> > I appreciate the authors’ detailed response and additional results. The clarification regarding the choice of flow matching and the corrected ablation table have addressed my previous concerns. The explanation of the soft VLM-policy connection is also clearer now. It is nice to have more details of this connection in the final manuscript.  With these points clarified, and considering the authors’ responses to other reviewers, I believe the current version is self-contained. While minor flaws may still remain, I will raise my score accordingly.

---

> > > ### Author Response · Authors · 2026-04-03
> > >
> > > We sincerely thank you for your positive feedback and constructive suggestions. All revisions agreed upon during the rebuttal period will be incorporated into our revised version.

---

### Official Review · Reviewer_91KS · 2026-03-10

**Soundness:** 3
**Presentation:** 3
**Significance:** 3
**Originality:** 2
**Overall Recommendation:** 4
**Confidence:** 5

**Summary:**

The paper revisits the fragmented design space of Vision–Language–Action (VLA) models and proposes a unified evaluation framework and a set of empirically validated recipes that substantially improve VLA performance and robustness. Through systematic ablations across foundational VLM–policy coupling, perception inputs (multi‑view vision, proprioception), and action modeling objectives (chunking, flow matching, frequency-domain auxiliary losses), the authors distill 12 findings and assemble them into a compact model, VLANeXt, which attains strong results on LIBERO and notably improved robustness on LIBERO‑plus, with promising real‑robot experiments.

**Compliance With Llm Reviewing Policy:**

Affirmed.

**Final Justification:**

The author's response addressed my questions and also supplemented the experimental details that I was most concerned about.

**Key Questions For Authors:**

1. Please detail the frequency-domain loss: transform length and windowing, whether it is applied per-action dimension independently, weighting across frequencies, and sensitivity to the 0.1–0.2 coefficient. Does it remain beneficial for non-smooth or contact-rich tasks?
2. Ablations are primarily on the Spatial suite. Which of the key findings (e.g., proprioception to VLM, soft coupling, action chunking) replicate on Object/Goal/Long in controlled ablations?

**Limitations:**

Apart from the relatively basic architectural ablations, most of the improved solutions on LIBERO-Plus show insignificant performance gains and introduce additional cumbersome designs. The authors should explicitly point out these limitations.

**Strengths And Weaknesses:**

Strengths:
1. A coherent, end-to-end empirical study that disentangles often conflated design choices in VLAs (VLM–policy interface, proprioception fusion locus, action objectives), culminating in a practical recipe.
2. Introducing a lightweight frequency-domain auxiliary loss for action forecasting is simple, principled from a time-series perspective, and effective with negligible overhead.
3. Comparisons against a wide set of prior VLA baselines on both LIBERO and LIBERO‑plus, plus a small but meaningful real-world evaluation (single-arm and bimanual tasks).
4. Clear organization around three design axes (foundational components, perception essentials, action modeling perspectives), with helpful schematic diagrams clarifying interfaces and modules.

Weaknesses:
1. Many design elements are incremental or adaptations of existing ideas (e.g., flow matching heads, MetaQuery‑like policy modules, multi-view, proprioception fusion), with moderate novelty; the main contribution is empirical consolidation rather than a fundamentally new modeling principle. The choices in most of these modules did not yield results that differed from or were counterintuitive relative to prior literature.
2. The choice to adopt flow matching despite regression and DDIM slightly outperforming it on LIBERO-Plus evaluation is under-justified, and the conditions favoring each remain under-explored.
3. Run‑time, latency, and compute/training budgets (pretraining/finetuning) are not systematically reported; practical throughput is critical for robot deployment and fair baseline comparison.

---

> ### Author Rebuttal · Authors · 2026-03-31
>
> **Q1: Novelty of the paper**
>
> **A1:** We believe our work offers meaningful novelty and contributes to the academic community in several ways.
>
> First, we investigated **several design aspects that remain unresolved** in the VLA literature.
>
> 1. **Use of proprioception**: many VLAs omit proprioception inputs, and some prior works [1] suggest they may not improve performance. In contrast, we show that incorporating proprioception into the VLM improves action learning.
>
> 2. **VLM-policy connection**: existing approaches vary widely (loose coupling, tight coupling, and no separate policy). We provide a controlled comparison and find that tight connections slightly outperform the loose ones, and both are superior to designs without a separate policy. We further introduce a soft connection that improves upon the tight variant.
>
> 3. **Role of video inputs**: most VLAs rely on single-frame inputs, and the benefits of temporal inputs remain unclear. Our results suggest that na\""{i}vely incorporating temporal inputs can be redundant and may even degrade performance.
>
> 4. **Frequency-domain loss**: this is an unexplored direction (only brefily considered in FAST[2] for a different purpose). We study this design choice and propose a new practice to this setting.
>
> Second, given the fragmented nature of current VLA research, we believe our work provides a study of key design choices, offering **clearer guidance** to the field. In this sense, our work serves both as a **unifying perspective** on existing approaches and as a **strong baseline** for future exploration.
>
> ---
>
> **Q2: Trend of different continuous losses**
>
> **A2:** From our observation, when the performance is moderate, regression performs better; when the performance gets higher, flow matching gets ahead since it can represent precise control signals, as seen below. And DDIM performs similarly to flow matching in most cases.
>
> | Model | LIBERO-Spatial | LIBERO-plus-Spatial |
> | --- | --- | --- |
> | Moderate baseline w. regression | 85.4 | 48.4 |
> | Moderate baseline w. FM | 80.0 | 45.0 |
> | VLANeXt w. regression | 97.2 | 92.4 |
> | VLANeXt | 99.0 | 92.8 |
>
> Moreover, our work not only outlines a practical roadmap for designing strong VLAs, but is also grounded in extensive empirical exploration. In the whole process, we try **more than 400 distinct combinations** in different design spaces, and the roadmap is just a distillation from these explorations. We believe our recipe is robust and reliable.
>
> ---
>
> **Q3: Speed of the method**
>
> **A3:** We test the training and inference speed of VLANeXt. As seen in the table below, VLANeXt can **train fast**, and **support real-time inference**. All the tests are conducted in an 80GB A800 machine, with a 12-core CPU and 120 GB main memory.
>
> | Model | Pretraining |  Franka Finetuning | Aloha Finetuning |  LIBERO Finetuning | Inference|
> | --- | --- | --- | --- | --- | --- |
> | VLANeXt |238 h | 62 h | 75 h | 23 h| 26 FPS|
>
> ---
>
> **Q4: Details about frequency loss**
>
> **A4:** The frequency loss is applied per-action dimension independently. We increase the weight of low frequency since the high frequency is noisy. The performance is not sensitive to the coefficient, and larger coefficients will be slightly better.
>
> ---
>
> **Q5: Performance for contact-rich task**
>
> **A5:** We further conduct a contact-rich task, **cooking**, in real world. This task involves a range of skills such as repeated pick-and-place movements, sauce addition, and stir-frying. As seen in the table below, our model can get **good performance**.
>
> | Model | Cooking Success Rate |
> | --- | --- |
> | VLANeXt | 5/10 |
>
> ---
>
> **Q6: Ablations in other settings**
>
> **A6:** We further conduct a series of ablations in the other suites, and also in the real world. The results can be seen below. Our finding is that the ablation results **are consistent with** the results in the Spatial suite.
>
> | Model | LIBERO-Spatial| LIBERO-Object | LIBERO-Goal | LIBERO-Long | LIBERO-avg | LIBERO-Plus-Spatial| LIBERO-Plus-Object | LIBERO-Plus-Goal | LIBERO-Plus-Long |LIBERO-Plus-avg|
> | --- | --- | --- | --- | --- | --- | --- | --- | --- | --- | --- |
> | VLANeXt - frequency - proprioception - multiview - soft | 90.0 | 31.2 | 93.6 | 83.4 | 74.6 | 53.7 | 38.2 | 36.7 | 50.9 | 44.9 |
> | VLANeXt - frequency - proprioception - multiview | 91.8 | 61.8 | 94.8 | 90.2 | 84.7 | 56.2 | 41.4 | 47.3 | 54.0 | 49.7 |
> | VLANeXt - frequency - proprioception | 97.6 | 98.8 | 96.0 | 93.8 | 96.6 | 80.5 | 74.4 | 69.0 | 68.1 | 73.0 |
> | VLANeXt - frequency | 98.0 | **99.4** | 95.6 | **95.0** | 97.0 | 87.7 | 77.9 | 71.3 | 71.4 | 77.1 |
> | VLANeXt | **99.0** | 99.2 | **96.6** | 94.6 | **97.4** | **92.8** | **82.1** | **72.7** | **72.8** | **80.1** |
>
> | Model | Real-world clean table |
> | --- | --- |
> | VLANeXt wo. action chunk | 1/10 |
> | VLANeXt wo. multiview | 6/10 |
> | VLANeXt wo. soft connection | 5/10 |
> | VLANeXt wo. proprioception | 4/10 |
> | VLANeXt | **7/10** |
>
> [1] Zhao et al. Do You Need Proprioceptive... arXiv:2509.18644.

---

> > ### Author Rebuttal · Reviewer_91KS · 2026-04-03
> >
> > The author's response addressed my questions and also supplemented the experimental details that I was most concerned about.

---

> > > ### Author Response · Authors · 2026-04-04
> > >
> > > We sincerely thank you for your positive feedback and helpful suggestions. We are glad to see that most of your concerns have been addressed. We will incorporate the supplementary experiments presented here into the revised manuscript.

---

### Official Review · Reviewer_CyTD · 2026-03-16

**Soundness:** 3
**Presentation:** 3
**Significance:** 2
**Originality:** 2
**Overall Recommendation:** 3
**Confidence:** 4

**Summary:**

This paper proposes VLA training protocols to systematically explore and verify the effectiveness of several VLA design choices, including policy module design, action chunking, action learning objectives, VLM backbones, VLM–policy connections, and perception input conditions.

**Compliance With Llm Reviewing Policy:**

Affirmed.

**Final Justification:**

While some concerns have been addressed, the main concerns regarding the contribution remain unresolved. Therefore, I maintain my score.

**Key Questions For Authors:**

1. Were there any non-trivial challenges in integrating the different components into a single VLA system, such as optimization instability, component interference, or sensitivity to implementation details?

**Limitations:**

yes

**Strengths And Weaknesses:**

**Strengths**
1. This work provides useful practical guidance for future VLA research and development. While VLA systems involve many design choices, relatively few studies have systematically analyzed the impact of individual components and consolidated the findings within a unified framework. Moreover, integrating diverse techniques into a single high-performing model requires substantial technical effort, which adds to the practical value of this work.

2. The paper offers a relatively thorough empirical analysis of component-wise effects. The comparisons are conducted on both LIBERO and LIBERO-plus, covering VLA performance from multiple perspectives. This experimental setup helps build a more detailed understanding of how each component influences overall performance.

3. The final model demonstrates strong empirical performance. The proposed model consistently outperforms strong VLA baselines such as pi0 and OpenVLA-OFT across LIBERO, LIBERO-plus, and real-world evaluation settings, which makes the overall results compelling.


**Weaknesses**
1. The main limitation of the paper is that its novelty appears more empirical and integrative than fundamentally technical. Overall, the paper reads closer to a technical report that consolidates and validates existing design insights than to a work introducing a clearly new modeling or learning method. In particular, several explored dimensions seem closely related to directions that have already been discussed in prior work. For example, action chunking horizon, camera view horizon, and time-series forecasting perspectives correspond to previously studied design choices; proprioception conditioning appears closer to an engineering design decision; and the usefulness of world modeling has already been supported by prior works such as DUST and FLARE.  This limits the contribution of the paper.

2. The other limitation is that all component-wise evaluations are conducted only within the VLANeXt framework. As a result, it remains somewhat unclear whether the conclusions drawn from these comparisons should be interpreted as broadly applicable design principles or as findings that are particularly well suited to this specific system configuration. Additional validation under other VLA setups would help clarify the broader applicability of these findings, which is currently somewhat limited.

---

> ### Author Rebuttal · Authors · 2026-03-31
>
> **Q1: The novelty of the paper**
>
> **A1:** We believe our work offers meaningful novelty and contributes to the academic community in several ways.
>
> First, we investigated **several design aspects that remain unresolved** in the VLA literature.
>
> 1. **Use of proprioception**: many VLAs omit proprioception inputs, and some prior works [1] suggest they may not improve performance. In contrast, we show that incorporating proprioception into the VLM improves action learning.
>
> 2. **VLM-policy connection**: existing approaches vary widely (loose coupling, tight coupling, and no separate policy). We provide a controlled comparison and find that tight connections slightly outperform the loose ones, and both are superior to designs without a separate policy. We further introduce a soft connection that improves upon the tight variant.
>
> 3. **Role of video inputs**: most VLAs rely on single-frame inputs, and the benefits of temporal inputs remain unclear. Our results suggest that na\""{i}vely incorporating temporal inputs can be redundant and may even degrade performance.
>
> 4. **Frequency-domain loss**: this is an unexplored direction (only brefily considered in FAST[2] for a different purpose). We study this design choice and propose a new practice to this setting.
>
> Second, given the fragmented nature of current VLA research, we believe our work provides a study of key design choices, offering **clearer guidance** to the field. In this sense, our work serves both as a **unifying perspective** on existing approaches and as a **strong baseline** for future exploration.
>
> ---
>
> **Q2: The VLA setup**
>
> **A2:** We believe our VLANeXt framework has **included and explored most of the setups** in previous VLAs. We would like to elaborate on this in the following aspects.
>
> 1. We start from the setup similar to RT2[3], then gradually add new designs, **including the key setups** for different VLAs (VLM choices, VLM-policy connections, and loss), to form the VLANeXt. We believe different key VLA setups can be combinations inside our framework.
>
> 2. Our work not only outlines a practical roadmap for designing strong VLAs, but is also grounded in extensive empirical exploration. In the whole process, we evaluated **more than 400 distinct combinations** across different design choices and engineering settings, and the proposed roadmap can be viewed as a distillation version of these explorations. We therefore believe that our study has already covered a broad range of setups and that the resulting conclusions are both robust and reliable.
>
> 3. To directly address your concern, we further conduct ablations in other suites (object, goal, and 10) in LIBERO, and also in the real world. It can be seen from the following table that the **conclusions are consistent** with our ablations in the Spatial suite in the main paper.
>
> | Model | LIBERO-Spatial| LIBERO-Object | LIBERO-Goal | LIBERO-Long | LIBERO-avg | LIBERO-Plus-Spatial| LIBERO-Plus-Object | LIBERO-Plus-Goal | LIBERO-Plus-Long |LIBERO-Plus-avg|
> | --- | --- | --- | --- | --- | --- | --- | --- | --- | --- | --- |
> | VLANeXt - frequency - proprioception - multiview - soft | 90.0 | 31.2 | 93.6 | 83.4 | 74.6 | 53.7 | 38.2 | 36.7 | 50.9 | 44.9 |
> | VLANeXt - frequency - proprioception - multiview | 91.8 | 61.8 | 94.8 | 90.2 | 84.7 | 56.2 | 41.4 | 47.3 | 54.0 | 49.7 |
> | VLANeXt - frequency - proprioception | 97.6 | 98.8 | 96.0 | 93.8 | 96.6 | 80.5 | 74.4 | 69.0 | 68.1 | 73.0 |
> | VLANeXt - frequency | 98.0 | **99.4** | 95.6 | **95.0** | 97.0 | 87.7 | 77.9 | 71.3 | 71.4 | 77.1 |
> | VLANeXt | **99.0** | 99.2 | **96.6** | 94.6 | **97.4** | **92.8** | **82.1** | **72.7** | **72.8** | **80.1** |
>
> | Model | Real-world clean table |
> | --- | --- |
> | VLANeXt wo. action chunk | 1/10 |
> | VLANeXt wo. multiview | 6/10 |
> | VLANeXt wo. soft connection | 5/10 |
> | VLANeXt wo. proprioception | 4/10 |
> | VLANeXt | **7/10** |
>
> ---
>
> **Q3: The non-trivial challenges in our explorations**
>
> **A3:** One non-trivial challenge is how to explore the VLA design spaces comprehensively, rather than just a straight line, to erase the interference of component combinations, sensitivity, and instability, as you proposed. To achieve this, we tried **more than 400 distinct experiments** to distill this useful recipe to the community.
>
> [1] Zhao J, et al. Do You Need Proprioceptive States in Visuomotor Policies?. arXiv:2509.18644, 2025.
>
> [2] Pertsch K, et al. Fast: Efficient action tokenization for vision-language-action models. arXiv:2501.09747, 2025.
>
> [3] Zitkovich B, et al. Rt-2: Vision-language-action models transfer web knowledge to robotic control. CoRL. 2023: 2165-2183.

---

> > ### Author Rebuttal · Reviewer_CyTD · 2026-04-03
> >
> > Setting aside the novelty-related aspects, I find that the remaining components are either not new or provide limited additional contribution:
> >
> >  - VLM-policy connection: The exploration of VLM-policy adapters does not differ significantly from prior work such as VLA-Adapter [1], which already investigates various connection strategies (e.g., loose, tight, and soft connections) and further develops additional methods.
> >  - Role of video inputs: It is already well known that incorporating video into imitation learning can sometimes degrade performance [2].
> >  - Frequency-domain loss: As mentioned, this is a commonly used technique in the tokenizer literature, including works such as FAST and FASTer [3]. Its effectiveness is relatively well established, making it more of a technical improvement than a conceptual contribution.
> >  - Use of proprioception: While the direct comparison is somewhat new, it remains largely a technical contribution and is not sufficient on its own to constitute a strong novelty claim.
> >
> >
> > Therefore, I still consider the overall contribution of this paper to be largely a combination of existing ideas, and I maintain my score.
> >
> >
> > [1] Wang, Yihao, et al. "Vla-adapter: An effective paradigm for tiny-scale vision-language-action model." Proceedings of the AAAI conference on artificial intelligence. Vol. 40. No. 22. 2026.
> >
> > [2] De Haan, Pim, Dinesh Jayaraman, and Sergey Levine. "Causal confusion in imitation learning." Advances in neural information processing systems 32 (2019).
> >
> > [3] Liu, Yicheng, et al. "FASTer: Toward Powerful and Efficient Autoregressive Vision–Language–Action Models with Learnable Action Tokenizer and Block-wise Decoding." The Fourteenth International Conference on Learning Representations.

---

> > > ### Author Response · Authors · 2026-04-03
> > >
> > > Thank you very much for your feedback and for further discussion. However, we **respectfully disagree** with some of your opinions.
> > >
> > > First, our paper is designed as a **systematic** paper, aiming to rigorously offer a **clearer guidance** and an **unified perspective** to the fragmented VLA design space, not to invent incremental modifications to current VLAs. Moreover, this kind of systematic work is also highly **important and has strong contributions** for the field's development.
> > > 1. **From Origin to Modern**: Our exploration is complehensive. We distill a recipe from more than 400 distinct experimental explorations, which connects the VLA development from its origin [1] to a modern, strong VLA for the future exploration. _Reviewers NsfZ and Yhj1 also acknowledge the contributions of this complehensive recipe._
> > > 2. **VLA-version ConvNeXt [2]**: We hope our contribution can play a role similar to that of ConvNeXt [2], which, as a systematic study of vision models, marked an important milestone in the evolution of convolutional neural networks.
> > >
> > > Second, setting aside its strong systematic value, we believe our work is also **sufficiently innovative**.
> > > 1. Our usage of frequency loss is **fundamentally different from FAST [3] and its variant [4]**. Their use of frequency is to train an action tokenizer for classification, while ours is to serve as a multitask learning objective. Our usage is lightweight, plug-and-play, which doesn't need large-scale pretraining at first, and also has strong performance and generalizability. _Reviewers 91KS and Yhj1 also acknowledge the contribution of our frequency-domain loss design._
> > > 2. As you acknowledged, our exploration of proprioception conditioning provides new, actionable insights to the field.
> > > 3. Our conclusion of video input is based on **native video understanding VLM, not just a naive from-scratch policy [5]**. We observe that Qwen3-VL, which is already pretrained in large-scale video understanding data and has mastered video processing ability, still fails to distill useful information for action learning, which is also surprising to us.
> > > 4. We acknowledge the connection explorations in VLA-adapter [6], but their **connection exploration is different from ours**:
> > >
> > >     (i). **Design is different**: Our soft connection differs from the four variants explored in VLA-Adapter; instead, it is closer to an All-layer Raw + All-layer ActionQuery connection.
> > >
> > >     (ii). **Exploration domains are different:** The connection exploration in VLA-adapter focuses on different tight connection variants (connect the middle features from VLM to policy), while ours explores and fairly compares the previous mainstream connection strategies in VLA (not separate, loose connection, tight connection), and propose a new soft connection combining the ideas from the loose and tight strategies.
> > >
> > > Based on the above four points, we modestly believe the novelty of our work is sufficient.
> > >
> > > In summary, we believe our work not only makes **strong systematic contributions**, but is also **sufficiently innovative**.
> > >
> > > [1] Zitkovich B, Yu T, Xu S, et al. Rt-2: Vision-language-action models transfer web knowledge to robotic control. CoRL. 2023: 2165-2183.
> > >
> > > [2] Liu Z, Mao H, Wu C Y, et al. A convnet for the 2020s. CVPR. 2022: 11976-11986.
> > >
> > > [3] Pertsch K, Stachowicz K, Ichter B, et al. Fast: Efficient action tokenization for vision-language-action models. arXiv preprint arXiv:2501.09747, 2025.
> > >
> > > [4] Liu Y, Zhang S, Dong Z, et al. FASTer: Toward Efficient Autoregressive Vision Language Action Modeling via Neural Action Tokenization[J]. arXiv preprint arXiv:2512.04952, 2025.
> > >
> > > [5] De Haan P, Jayaraman D, Levine S. Causal confusion in imitation learning. NeurIPS, 2019, 32.
> > >
> > > [6] Wang Y, Ding P, Li L, et al. Vla-adapter: An effective paradigm for tiny-scale vision-language-action model. AAAI. 2026, 40(22): 18638-18646.

---

### Official Review · Reviewer_NsfZ · 2026-03-18

**Soundness:** 3
**Presentation:** 3
**Significance:** 2
**Originality:** 2
**Overall Recommendation:** 4
**Confidence:** 4

**Summary:**

This paper studies the design choices of Vision–Language–Action (VLA) models and aims to identify which architectural and training decisions are critical for performance. Starting from a VLA baseline, the authors systematically explore design choices along three axes: foundational components, perception, and action modeling.

By combining a set of empirically effective design choices, the paper proposes VLANeXt, which achieves state-of-the-art performance on both LIBERO and LIBERO-plus, and demonstrates promising real-world results.

**Compliance With Llm Reviewing Policy:**

Affirmed.

**Final Justification:**

The authors have addressed most of my concerns. I will maintain my positive score.

**Key Questions For Authors:**

1. Can the authors provide deeper analysis to clarify why the soft connection outperforms loose and tight connections?

2. How does the proposed frequency-domain loss compare to existing approaches for structured action modeling, such as methods like FAST[2]?

3. For the real-world experiments, how sensitive are the results to the proposed design choices?

4. How well will the proposed design choices generalize beyond the relatively simple tasks currently evaluated on real robots?

[2] Pertsch, Karl, et al. "Fast: Efficient action tokenization for vision-language-action models." arXiv preprint arXiv:2501.09747 (2025).

**Limitations:**

Yes

**Strengths And Weaknesses:**

Strengths:
- The paper conducts careful and controlled ablations across a wide range of design choices.
- It provides a practical and valuable training recipe, which can serve as a useful reference for future VLA research.
- VLANeXt achieves strong performance on LIBERO and LIBERO-plus, despite using a relatively small backbone.

Weaknesses:
- The real-world evaluation is limited in scope, with only 4 relatively simple tasks and 20 rollouts per task. Since the paper positions itself as a general training recipe for VLA models, more comprehensive real-world evaluation (e.g., more diverse tasks, longer-horizon behaviors, or more challenging settings) would better support the claims.
- While running full ablations in real-world settings is understandably expensive, it would be important to include at least some critical ablations in real-world experiments, especially for key design choices that show significant gains in simulation (e.g., soft connection, or proprioception conditioning).
- The baselines do not include more recent and stronger VLA baselines (e.g., π0.5[1] rather than π0).

[1] Intelligence, Physical, et al. "$\pi_ {0.5} $: a Vision-Language-Action Model with Open-World Generalization." arXiv preprint arXiv:2504.16054 (2025).

---

> ### Author Rebuttal · Authors · 2026-03-31
>
> **Q1: Long-horizon real-world experiments**
>
> **A1:** We further evaluate our model on a long-horizon real-world task, **cooking**, which involves a range of skills such as repeated pick-and-place movements, sauce addition, and stir-frying. As shown in the table below, our model maintains **strong performance** in this long-horizon setting.
>
> | Model | Cooking Success Rate |
> | --- | --- |
> | VLANeXt | 5/10 |
>
> ---
>
> **Q2: Ablations in the real world**
>
> **A2:** We conduct ablation studies on the real-world task of cleaning table, with results shown below. We find that the real-world ablation results **are consistent** with our findings from the simulation benchmark.
>
> | Model | Real-world clean table |
> | --- | --- |
> | VLANeXt wo. action chunk | 1/10 |
> | VLANeXt wo. multiview | 6/10 |
> | VLANeXt wo. soft connection | 5/10 |
> | VLANeXt wo. proprioception | 4/10 |
> | VLANeXt | **7/10** |
>
> ---
>
> **Q3: Comparison with pi0.5**
>
> **A3:** The comparative results against pi0.5 on the LIBERO benchmark are presented below, showing that our framework offers **better performance**.
>
> | Model | Spatial | Object | Goal | Long | Avg |
> | --- | --- | --- | --- | --- | --- |
> | pi0.5 | 98.8 | 98.2 | **98** | 92.4 | 96.9 |
> | VLANeXt | **99.0** |  **99.2** | 96.6 | **94.6** | **97.4** |
>
> ---
>
> **Q4: Analysis of Soft Connection**
>
> **A4:** Our understanding of the soft connection is two-fold:
>
> 1. **Better leverage of VLM features:** A layer-by-layer connection can better leverage the VLM's representations than a looser design that only links the policy with the metaqueries' final-layer features.
>
> 2. **Serves as a buffer:** The action output space differs considerably from the text output space in the VLMs. The soft connection adds learnable queries between the VLM and the policy before the layer-by-layer connection, serving as a buffer to bridge the gap between text and action spaces.
>
> The real-world ablation results above further support these observations.
>
> ---
>
> **Q5: Comparison with FAST**
>
> **A5:** We experimented with the FAST action tokenizer in our framework under several settings, including directly using the pretrained FAST tokenizer, training a FAST tokenizer on LIBERO, and varying the training hyperparameters. However, all of these attempts performed poorly, yielding less than a 15\% success rate on the LIBERO Spatial suite. Our current hypothesis is that the action patterns induced by FAST are difficult to learn effectively without large-scale pretraining.
>
> In the main paper, we compare our method against a baseline that uses the FAST action tokenizer, namely pi0-FAST. The results, summarized in the table below, show that our method achieves **better performance**.
>
> | Model | LIBERO-Avg | LIBERO-plus-Avg |
> | --- | --- | --- |
> | pi0-FAST | 85.5 | 61.6 |
> | VLANeXt | **97.4** |  **80.1** |
>
> ---
>
> **Q6: Sensitivity in the real world**
>
> **A6:** We conduct ablations of several key designs in the real-world setting (see A2). The conclusions are consistent with our findings from the simulation benchmark. Our preliminary experiments suggest that real-world training is more stable than that in simulation against training hyperparameters. We attribute this to the fact that VLMs are primarily pretrianed on real-world data, making them easier to adapt during real-world fine-tuning.
>
> ---
>
> **Q7: Generation beyond simple tasks in the real world**
>
> **A7:** We evaluate our model on a long-horizon real-world task, cooking (see A1). The results are comparable to those on simpler tasks (e.g., table cleaning and object lifting) reported in the main paper.

---

> > ### Author Rebuttal · Reviewer_NsfZ · 2026-04-04
> >
> > Thank you for your detailed rebuttal and the additional experimental results supporting your claims. The authors have addressed most of my concerns. I will maintain my positive score.

---

> > > ### Author Response · Authors · 2026-04-04
> > >
> > > Thank you for the positive evaluation and constructive comments. We are glad to have addressed your main concerns. The supplementary experiments provided here will be included in the revised paper.

---

### Decision · Program_Chairs · 2026-04-30

**Decision:**

Accept (regular)

**Comment:**

This paper received a mixed set of scores. While the reviewers praised the value of this paper as a well-written and well-executed systematic paper, reviewers also raises concerns about the novelty and the limited evaluation in the original draft. While I agree with the reviewer CyTD's concern in that many of observations themselves have also been made in the prior work, I'm more in agreement with major opinion in that the paper is a good systematic paper that can be helpful for future researchers and practitioners that will work with VLAs. Therefore, I recommend the acceptance of this paper.